# Developments in Exploring Fungal Secondary Metabolites as Antiviral Compounds and Advances in HIV-1 Inhibitor Screening Assays

**DOI:** 10.3390/v15051039

**Published:** 2023-04-23

**Authors:** Bruce Nzimande, John P. Makhwitine, Nompumelelo P. Mkhwanazi, Sizwe I. Ndlovu

**Affiliations:** 1Discipline of Medical Microbiology, School of Laboratory Medicine and Medical Sciences, Medical School, University of KwaZulu-Natal, Durban 4000, South Africa; 2HIV Pathogenesis Programme, Doris Duke Medical Research Institute, School of Laboratory Medicine and Medical Sciences, University of KwaZulu-Natal, Durban 4000, South Africa; 3Department of Biotechnology and Food Technology, Doornfontein Campus, University of Johannesburg, Johannesburg 2028, South Africa

**Keywords:** antiviral, endophytic fungi, human immunodeficiency virus-1, immunomodulation, secondary metabolites

## Abstract

The emergence of drug-resistant Human Immunodeficiency Virus-1 strains against anti-HIV therapies in the clinical pipeline, and the persistence of HIV in cellular reservoirs remains a significant concern. Therefore, there is a continuous need to discover and develop new, safer, and effective drugs targeting novel sites to combat HIV-1. The fungal species are gaining increasing attention as alternative sources of anti-HIV compounds or immunomodulators that can escape the current barriers to cure. Despite the potential of the fungal kingdom as a source for diverse chemistries that can yield novel HIV therapies, there are few comprehensive reports on the progress made thus far in the search for fungal species with the capacity to produce anti-HIV compounds. This review provides insights into the recent research developments on natural products produced by fungal species, particularly fungal endophytes exhibiting immunomodulatory or anti-HIV activities. In this study, we first explore currently existing therapies for various HIV-1 target sites. Then we assess the various activity assays developed for gauging antiviral activity production from microbial sources since they are crucial in the early screening phases for discovering novel anti-HIV compounds. Finally, we explore fungal secondary metabolites compounds that have been characterized at the structural level and demonstrate their potential as inhibitors of various HIV-1 target sites.

## 1. Introduction

The increasing number of people living with Human Immunodeficiency Virus (HIV) remains a significant global public health concern. According to UNAIDS report, more than 38 million people worldwide were living with HIV in the year 2021. About 20.6 million of these people living with HIV were in the Eastern and Southern Africa regions where HIV-1 is the predominant strain [1]. Approximately, 75.5% people infected with HIV globally are receiving antiretroviral therapy which indicate a remarkable progress [1]. The introduction of combination antiretroviral (cARV) therapies has reduced HIV-1 from a fatal disease to a manageable chronic condition owing to reduced probabilities of the disease progression to acquired immunodeficiency disease syndrome (AIDS) and thus resulting in a significant reduction to HIV-1-related morbidity and mortality [2]. Despite these gains, about 650,000 people in 2021 succumbed to AIDS-related illnesses. Although a large extent of these deaths could be a result of limited access to antiretroviral therapy in some regions, some other factors includes the limited therapeutic effect since these drugs do not cure HIV due to failure to eliminate latently integrated HIV proviral DNA in the host genome, which is capable of initiating replication to detectable levels once ARV treatment is interrupted [3]. Virological failure is reported in at least 20% of individuals receiving the first line of ARVs in low- and middle-income countries [4]. The success story of the existing antiretroviral regimens is hindered by the emergence of antiviral-resistant HIV-1 strains and adverse side effects leading to non-adherence [4,5]. For example, long-term use of ARTs comes with the risk of myopathy, lymphadepathy, cardiotoxic effects, and other hematologic disorders [6,7,8]. The repeated use of ARTs may cause the evolution of viral mutations in infected individuals leading to suppression of the bone marrow and high mitochondrial toxicity [9]. Moreover, current antiviral drugs cannot inhibit the virus from replicating in viral reservoirs. Human immunodeficiency virus continues to persist within latent cellular reservoirs, which can become reactivated at any time to produce an infectious virus [10,11]. New therapies are therefore required not only for HIV suppression but also for containing or eliminating HIV reservoirs.

One of the long-term research approaches that is gaining impetus is directed towards implementing small molecule screens of naturally derived compounds, specifically to identify compounds that are effective for provirus reactivation in combination with previously Food and Drugs Administration (FDA) approved drugs. Fungal species, especially endophytic fungi associated with medicinal plants are rich sources of novel and bioactive small molecular (secondary metabolites) compounds [12,13,14,15]. Although this review will focus on all fungal metabolites, endophytic fungi are of particular interest since they represent a largely unexplored niche. To date, no fungal metabolites have been approved as treatment for HIV-1, although there are several early screening studies suggesting fungal derived natural products to hold a high potential as anti-HIV therapies. Some of these studies have been extensively reviewed by Linnakoski et al. [15] and Roy [16]. This review intends to highlight structurally elucidated compounds derived from fungi/endophytic fungi as alternative HIV treatment strategies that have the potential to expand HIV-1 treatment options and escape existing treatment barriers. This review will explore the advances in developing antiviral assays, which are vital to uncovering antiviral agents from microbial sources. Furthermore, we propose comprehensive, whole replication antiviral screening assays as the most informative and unbiased approach for early drug development initiatives that will assist in charting a future direction in the search for antiviral compounds from microbial sources. Finally, we examine the potential of secondary metabolites derived from endophytic/fungi as future antiviral agents that can be used as potentiators, inhibitors, or latency-reversing agents to combat HIV.

## 2. The Biology of HIV and Currently Available Therapies for HIV Treatment

Human immunodeficiency virus infection begins when the viral glycoprotein gp120 binds to CD4+ receptors on the T helper cell [17]. Once attached, the CD4+ T cell and the HIV capsid fuse, allowing the viral RNA, reverse transcriptase, integrase, and protease to enter the cell [18,19]. Inside the host cell’s cytoplasm, the HIV RNA is converted into HIV DNA, which is facilitated by the reverse transcriptase enzyme [20]. The reverse transcriptase achieves this by transcribing viral RNA into dsDNA [17,21,22]. The viral DNA is transported into the host cell nucleus, which is then incorporated into the host’s genome with the help of the integrase enzyme [19].

Upon integrating the viral DNA into the host genome, the viral DNA can remain latent for many years [17]. In the absence of virus latency, the virus undergoes replication, utilizing the host machinery to produce long chains of viral proteins. The HIV proteins and the viral RNA are transported to the host cell’s surface, where they assemble into non-infectious particles [22,23]. After budding, the viral protease, mediates a proteolytic cleavage of the viral capsid, resulting in a mature infectious virus particle (virion) [22,24,25]. Current treatment requires that people living with HIV stay on antiretroviral therapy for life; otherwise, hidden HIV proviral DNA in cellular reservoirs will reactivate when treatment is interrupted.

There have been significant advances in the treatment, control, and prevention of HIV since its discovery in the 1980s [26]. In 1987, the Food and Drug Administration (FDA) approved its first drug, zidovudine (AZT), which initially failed in cancer screens, to treat HIV [27]. The synthesis of this drug was inspired by a naturally discovered thymidine from a marine sponge, *Tectitethya crypta*, known as sponge thymidine [28]. The discovery of AZT has paved the way for several classes of antiretroviral drugs currently on the market. The FDA classifies antiretroviral drugs for HIV infection into the following categories: nucleoside reverse transcriptase inhibitors (NRTIs), non-nucleoside reverse transcriptase inhibitors (NNRTIs), protease inhibitors (PIs), fusion inhibitors, entry inhibitors—CCR5 coreceptor antagonist and HIV integrase strand transfer inhibitors [29,30,31].

Highly active antiretroviral therapy (HAART) has significantly reduced the morbidity and mortality of HIV/AIDS. Antiretroviral therapy is currently recommended for all adults infected with HIV, and it has also reduced mother-to-child transmission to less than 1% [32]. Bhatti et al. [18] showed that taking anti-HIV medication alone is less effective than a combined ARV treatment regimen. A single-drug treatment allows the virus to mutate over time and build resistance to the medication. In combination therapy, more than one antiretroviral drug is administered concurrently to minimize the chances of resistance development. Recommendations for initial regimens include two nucleoside reverse transcriptase inhibitors, such as abacavir with lamivudine or tenofovir disoproxil fumarate, in combination with a third active ARV drug. The third drug can be from one of the three classes integrase strand transfer inhibitor (INSTI), a non-nucleoside reverse transcriptase inhibitor, or protease inhibitor (NNRTI) with a pharmacokinetic enhancer [33]. The first-line regimen currently in use is the combination of dolutegravir, lamivudine and tenofovir disoproxil fumarates. This is the combination of integrase and reverse transcriptase inhibitors. Despite the combination therapy, there is still a problem of HIV-1 drug resistance and the combination is also not effective in eradicating the virus in the latent reservoirs [34]. In fact, since the launch of potent ART, treatment recommendations have continued to prescribe regimens that include two NRTIs plus a third drug as preferred options [35]. The improved therapeutic and safety profile of treating all HIV-positive individuals as soon as a diagnosis is confirmed is paramount as it decreases mortality and increases life expectancy and the standard of treatment in antiretroviral therapy patients living with HIV [36].

The development of drug resistance to currently available antiretroviral drugs is a significant cause of treatment failure in HIV patients. The current antiretroviral therapies are also associated with adverse side effects and cell toxicity which often leads to non-adherence [37,38]. Resistance to NRTIs occurs through two mechanisms; (1) mutations resulting in reduced incorporation of the NRTI into the growing DNA chain and (2) enhanced removal of a drug from its attachment site at the end of the DNA chain [39]. These RT mutations allow ATP or pyrophosphate to bind adjacent to the bound nucleoside analogue at the active site. High-energy ATP or pyrophosphate can attack the bond that binds the drug to DNA, resulting in the drug’s liberation and termination of its effects. To combat antiviral resistance, there is an urgent need for the development of new anti-HIV drugs since approved ARVs are associated with various shortfalls such as toxicity and rapid emergence of resistant strains.

To further optimize therapeutic options and ensure a sustainable flow of new drugs with novel mechanisms of action, there is a need to ramp up early drug development initiatives. New antiviral drugs should meet specific criteria, such as not being susceptible to current resistance mechanisms. This can be achieved by targeting sites independent of the viral mutation development, such as those that target protein–protein interactions and host-mediated immune response to infection. In addition, current antiretroviral drugs only target the structural proteins of HIV. Thus, there is a need to develop novel antiretroviral therapies that will also target accessory proteins to achieve a comprehensive therapeutic approach. Advances have been made with small molecule metabolites, especially from microorganisms such as fungi [12,40]. Although these studies are in the early discovery phases, there are promising developments that call for more focus on optimizing antiviral screening assays and investment in developing hit compounds to lead compounds and eventually, clinical trials.

## 3. Advances in Antiviral Screening Assays

The development of innovative and highly sensitive activity assays for screening natural products with antimicrobial properties is a crucial step in identifying novel compounds [41]. The selection of an adequate bioassay and its validation is important in the initial stages of a drug development process (hit identification) for evaluating and prioritizing candidates for hit-to-lead optimization. Antiviral efficacy is difficult to determine since different viruses must be evaluated using different cell systems, making it impractical to devise a single assay for all viruses [42,43]. The use of high assay dosages, insufficient test controls, and the inaccurate collection of targets and endpoints represent only a couple of the main challenges facing antiviral screens. In addition to these factors, the effect of different reaction parameters also needs to be considered when designing antiviral screening experiments and interpreting output data [44]. The lack of standardization of methodologies contributes to widely contradictory outcomes, which presents a severe barrier to developing antiviral drugs from phenotypic screens of small microbial metabolites.

Several antiviral screening assays have been described for testing the antiviral efficacy of natural products, and these can be adapted at medium-to-high throughput formats [45]. Screening of inhibitor molecules using purified, or cell-surface-displayed HIV-1 proteins has been demonstrated with the major HIV-1 enzymes (reverse transcriptase, protease and integrase) to provide rapid results. Moreover, these single-protein assays are under research and development for rapid screening of inhibitors against accessory proteins such a Vpu or Vpr [46]. These assays are easy to perform and do not require highly specialized laboratory facilities [47]. However, single-target systems using purified, or cell-surface-expressed HIV-1 protein in vitro often leads to off-targets and cannot be replicated in cell culture systems [46,47]. In addition, inhibitor screening using single target HIV-1 protein requires each target assay to be set up separately which is costly and time-consuming. Ideally, a preliminary assay (e.g., a full replication assay or in silico target binding assay) should be conducted to predict the single target HIV-1 protein to choose for screening which can also prevent wasting costly resources [46].

On the other hand, systematic cell culture-based assays have gained increasing interest since they use chimeric pseudovirus particles that can be performed in level 2 biosafety laboratories and provide reliable results. The pseudovirus is constructed using a reporter gene such as luciferase and envelope proteins from a pathogenic virus-like HIV-1. These are displayed on the surface of a benign carrier virus such as Vesicular Stomatitis virus [48]. Mononuclear cells are infected with a chimeric pseudovirus and incubated with different concentrations of compounds [49]. The viral replication activates the reporter gene, and the activity can be measured in a calorimetric or fluorometric assay [50]. The EC50 values (reciprocal dilution needed to avoid virus-induced cytolysis by 50%) are used to express viral activity. Luciferase assay (measuring luciferase activity) is one example of a more rapid single-round infectivity assay that is widely used to assess the activity of compounds, including those from microbial natural products (e.g., fungal extracts) in HIV-1 strains [49].

An alternative single-round phenotypic assay replaces the luciferase indicator gene with the enhanced green fluorescent protein (GFP) gene and uses the HIV-1 CXCR4 or CCR5 tropic envelope instead of the MLV envelope [50,51]. Production of the virus is carried out similarly in HEK 293T cells, but infection is done in primary CD4+ T cells, the assay target cells of HIV-1 in vivo, rather than transformed cell lines. Infecting primary CD4+ T cells has the advantage of allowing measurement of drug inhibition by HIV-1 entry inhibitors such as enfuvirtide. In contrast, cell lines such as 293T must be transfected with the CD4+ T cells and the CXCR4 or CCR5 coreceptors to allow for HIV-1 envelope-mediated entry. Infectivity (number of GFP-expressing cells) can then be measured using flow cytometric quantitation [51,52].

The cell-culture-based assays have been expanded to cover other less-targeted viral proteins of HIV involved in forming viral structures and replications [53]. The p24 is a specific HIV structural protein that forms most of the virus core. This protein is secreted in the blood serum of an infected person and is therefore also recognized as an early biomarker of viral infection [54,55]. A study was undertaken to evaluate the anti-HIV activities of the *Morus alba* plant extract, mulberroside C, and extracts of endophytes isolated from the *M. alba* plant sample using in vitro assays [56]. In this study, peripheral blood mononuclear cells (PBMC) were infected with HIV, and the anti-HIV activity of mulberroside C was evaluated using HIV-1 p24 enzyme-linked immunosorbent assay (ELISA). While cell-culture-based viral inhibitor screening assays provide reliable results and any target can be engineered into the viral reporter gene, they are still limited to incorporating one target at a time which is laborious and costly [56]. The other challenge for single-target assays in determining viral inhibition is that they cannot evaluate the virus’s efficient replication, genetic diversity, and complex invasion strategy.

Recently, unbiased HIV-1 screening assays representing complete replication stages have been reported. These assays include the time of addition (TOA) assay reported by Daelemans et al. [47]. This assay is based on the principle that viral replication occurs in sequential stages that can be timed. The compound target site can be predicted based on its relative position in comparison with the reference drug used as the training set on the time scale. Another HIV-1 complete replication assay incorporating an accessory protein, Vif, was reported [57,58]. A summary of medium-to-high throughput screening assays that have been used successfully in screening small molecular metabolites is provided in Figure 1 [54,55,56,57,58,59,60,61,62,63,64,65,66]. Developing these replication assays to represent all possible target sites in the HIV-1 genome will allow for rapid and unbiased screening of natural microbial products against various targets and save costs and time for hit identification [64,65]. Furthermore, full replication assays will also provide insight into protein–protein interactions, and thus increase the early small molecular compound screening capacity for future viral inhibitors [61,66,67,68,69,70,71,72,73,74,75,76].

## 4. Advances in Developing Fungal-Derived Small Molecular Inhibitors for Major HIV-1 Target Sites

The fungal kingdom presents a great source of small therapeutic compounds with antibacterial, antifungal, and antiviral properties [77]. Unlike plant-derived secondary metabolites, fungal secondary metabolites can be scaled-up at a reasonable cost and time due to the availability of well-established industrial production parameters using fungal species [16]. Fungi from unique ecological niches, such as those found in the inner tissues of healthy plants, known as endophytic fungi, have been of great interest over the last two decades given their potential for use in drug development. The endophytes often reside within their host plants without causing any harm to the host plant [78]. In addition, secondary metabolites from microbial sources in novel niches, such as endophytic fungi, present compounds with structural complexities that cannot be matched by synthetic molecules, making them excellent candidates for new drug development. Fungal secondary metabolites often exhibit novel mechanisms of action that can escape already established mechanisms of drug resistance by microbial pathogens [79,80]. While the exploration of fungal compounds exhibiting antiviral activities is an emerging field, there have been several compounds revealed to possess high potential as viral inhibitors in several early screening programs as reviewed by Roy et al. [16] and Linnakoski et al. [15]. At this stage, there are currently no anti-HIV compounds derived from fungal sources in the clinical pipeline and almost all compounds are at an early discovery stage (hit identification). Although fungi have been widely explored as a source of active secondary metabolites, recent genome data suggest that what is currently known represents only a small perceptible part of a much larger reservoir. There is more biosynthetic potential that has not been explored [81,82,83,84]. Thus, there is plenty of scope for finding many potential anti-HIV drugs by exploring novel microbial species such as endophytic fungi.

## 5. Fungal Compounds as HIV-1 Cell-Surface Receptor Attachment Inhibitors

The fungal kingdom has been shown as a potential source of several bioactive compounds targeting the HIV-1 cell surface receptor attachment proteins. For instance, two novel chemokine receptor (CCR-5) inhibitors, Sch 210971 and Sch 210972, were isolated from an endophytic fungus, *Chaetomium globosum Kunze* 1705 with a potent in vitro activity of 79 nM [83]. It is well established that the chemokine receptors CCR5 and CXCR4 are required as coreceptors for binding gp120 and CD4+ T cells [84,85,86]. Inhibition of such binding can prevent viral entry into the host cell and prevent replication, representing a highly effective alternative target for HIV therapy. Several other anti-HIV-1 compounds derived from fungi targeting the gp120–CD4+ binding have been discovered, including isochromophilones I and II obtained from *Penicillium multicolor FO-2338*. These compounds disrupt gp120 and CD4+ interaction at 6.6 and 3.9 µM in in vitro assays, respectively, and prevent HIV-1 entry into the host cell [85,86,87]. Other isochromophilones A–F isolated from the Marine Mangrove endophytic fungi *Diaporthe* sp. SCSIO 41011 were reported to be cytotoxic and induced apoptosis in three different cell lines [88]. This finding emphasizes the importance of cytotoxicity screening of the fungal secondary metabolites before antiviral testing. Bioassay-guided isolation of the fungus, *Emericella aurantiobrunnea* led to the discovery and subsequent purification of a compound known as variecolin. Variecolin was observed to compete with macrophage inflammatory protein (MIP)-1R for binding into human CCR5 with an IC50 value of 9 µM [87]. These results on hit identification provide evidence of the potential of fungal-derived secondary metabolites in inhibiting HIV entry and replication. However, further studies of these hit compounds would offer more insight into their mechanisms and potential for further development into potential therapeutic agents.

## 6. Fungal Compounds as Reverse Transcriptase Inhibitors

Fungal natural products have also been explored as a source of reverse transcriptase inhibitors. The reverse transcriptase enzyme in HIV-1 infection plays a crucial role in viral replication by generating complementary DNA from an RNA template early in the HIV-1 life cycle. The role of reverse transcriptase makes it one of the most crucial targets for potential anti-HIV therapy [89]. Several other fungal species produce secondary metabolites that inhibit viral reverse transcription. This includes a study by Bashyal et al. [90], altertoxins extracted from a fungus, *Alternaria tenuissima* were found to inhibit the reverse transcriptase enzyme from transcribing HIV-1 RNA into DNA. Moreover, Melappa et al. [91] reported total coumarins from the crude extract of an endophytic fungus, *Alternaria* species to inhibit reverse transcriptase enzyme in a single protein in vitro colorimetric assay. The limitations of this study included that they only reported total coumarins and no purification of the active compounds was reported. However, the results of these studies provide some indication on the potential of fungi as a source of reverse transcriptase inhibitors. Additional research efforts are needed to validate this assertion.

## 7. Fungal Compounds as Integrase Inhibitors

The other important target in the HIV life cycle is the integration of the HIV viral DNA into the nucleus of the human genome facilitated by the enzyme integrase. Briefly, integration occurs via two catalytic actions, i.e., 3′-end processing and strand transfer [92]. The challenge with the current antiretroviral drugs is that HIV-1 has developed resistance to approved protease inhibitors primarily due to amino acid mutations within or proximal to the drug’s catalytic binding site [92].

The initiatives to discover naturally derived integrase inhibitors have started to indicate some successes. For instance, Singh et al. [93] showed that *Penicillium chrysogenum* produced xanthoviridicatins E and F, showing potent inhibition of HIV-1 integrase enzyme at highly effective concentrations of 6 and 5 µM, respectively. Both compounds inhibited the cleavage process of HIV-1 integrase. In the same year, a novel polyketide, cytosporic acid extracted from the endophytic fungus, *Cytospora* species, showed an in vitro HIV-1 strand transfer inhibition with IC50 of 20 μM [94]. Two other structurally similar fungal compounds, australifungin and australifungol, were evaluated, and only the former exhibited integrase inhibition activity with the same activity as cytosporic acid [94]. Australifungin was first discovered in fungal cultures of *Sporormiella australis* as an inhibitor of sphinganine N-acyl transferase, a mechanism common to mycotoxins such as fumonisin B [95]. The activity of both cytosporic acid and australifungin was attributed to the presence of a β-keto aldehyde group and a carboxyl group, respectively. Since both these compounds exhibited weak activities, their activities could further be improved by derivatization of the active pharmacophore to develop drug-like compounds [96]. Other success indicators for fungal-derived HIV-1 integrase inhibitor include the discovery of a phenalenone (atrovenetinone methyl acetal) from *Penicillium* species [97]. The authors compared this compound to phenalenones that they had previously isolated from fungal cultures, erabulenol B and funalenone, where the latter exhibited an intense integrase inhibition activity at 1.7 μM. Melappa et al. [91] also reported that the endophytic fungi, *Trichoderma harzanium* and *Alternaria* species isolated from a plant, *Calophyllum inophyllum* produced coumarins that showed inhibition of HIV-1 integrase. This study did not identify the coumarin derivative that was responsible for HIV-1 integrase inhibition. Coumestan-type isolated from *the Calophyllum inophylum* demonstrated anti-HIV-1 activity against integrase [98,99].

## 8. Fungal Compounds as Protease Inhibitors

Fungal protease inhibitors also reveal strong potential as future candidates in developing antiviral drugs. Singh et al. [100] reported hinnuliquinone as a potent inhibitor of the HIV-1 protease from an unidentified endophytic fungus inhabiting the leaves of Oak trees (*Quercus coccifera*) [100]. Another study used novel triterpenoids, gadoteridol F, 20-hydroxlucidenic acid N, ganoderic acid GC-2 and 20(21)-dehydrolucidenic acid N which were isolated from fungus, *Ganordema sinense* exhibited HIV protease-inhibitory activity at IC50 22, 25—30 and 48 μM, respectively [101]. Recently, Vora et al. [56] performed a targeted investigation of mulberroside C based on in silico observations that showed high-affinity binding to various target sites in the HIV cycle, especially as a protease inhibitor. Furthermore, out of the studied endophytic isolates, the extracts of MaF04C identified as *Phoma* species showed the most significant viral inhibition with IC50 of 8.19 ng/mL and less than 0.001 μg/mL in a TZM-bl cell-based β-glucosidase assay and HIV-1 p24 luciferase-based assay, respectively [56]. They further determined that MaF04C and MaF01SG (identified as *Chaetomium* species) had the highest relative protease inhibition profile, thus confirming the suggested protease inhibitory properties observed in the in silico studies [56]. These bioactive compounds have not proceeded to clinical trials or been validated beyond the early screening stages. Therefore, laboratory assays and in vivo tests are needed to fully understand the bioactivity, toxicity, and pharmacokinetic profile of the viral protease inhibitors produced by these endophytic fungi.

Ryang et al. [99] reported that paclitaxel could inhibit HIV-1 protease activity like the positive control, pepstatin A, in their in vitro experiment. A combination of paclitaxel and one already existing protease inhibitor (indinavir, nelfinavir, or combinations of these agents) at recommended dosages and schedules have been used to treat patients with HIV-associated Kaposi’s Sarcoma without enhancing toxicity [100,101,102,103,104]. However, patient conditions for paclitaxel applications should be considered because of its side effects on bone marrow suppression. Ryang et al. [99] indicate that fungal-derived compounds can also be used to potentiate existing HIV inhibitors in combination therapy which is a proven strategy for escaping resistance development.

## 9. Advances in Developing Fungal-Derived Small Molecular Inhibitors for New HIV-1 Target Sites

Despite the cART being potent and life-prolonging, a significant limitation is that it is not curative and does not eradicate HIV-1 infection since treatment interruption inevitably results in a rapid rebound of viremia [105]. The rebound of HIV is due to the presence of HIV reservoirs, mainly in the latently infected resting CD4+ memory T cells and myeloid cells such as macrophages and microglia that are difficult to target by cART or immune effector mechanisms [106,107,108,109]. HIV latency is defined as a reversibly, non-productive state of infection of individual cells that retain the capacity to produce infectious virus particles but allow the virus to evade the host’s immune response [110]. Latently infected cells contain stably integrated, replication-competent proviruses repressed at transcriptional and post-transcriptional levels by many silencing mechanisms [111,112].

Advances have been made in the discovery of fungal latency reversing agents (LRAs) with promising latency-reversing activities. Niu et al. [107] identified 12 eutypellazine (A–L) compounds with antiviral activity from a fungus, *Eutypella* sp. MCCC 3A00281. These compounds inhibited viral replication in the 293T HIV cell model with low toxicity. The compound eutypellazine J (80 μM) showed latency reactivation of HIV-1 in J-latA2 cells in a dose-dependent manner. The observed latency-reversing activity was comparable to that of the positive controls, prostratin (5 μM) and SAHA (suberoylanilide hydroxamic acid, 2.5 μM) [111]. Stoszko et al. [108] recently isolated gliotoxin from a fungal secondary metabolite library and showed this compound to exhibit latency-reversing properties. The compound was derived from *Aspergillus fumigatus* CBS 100074, and this was shown to disrupt 7SK small nuclear ribonucleoprotein (snRNP), resulting in the release of the positive transcription elongation factor B (P-TEFb) and subsequent reversal of HIV-1 latency [108]. The latency-reversing mechanism shown by gliotoxin is like that of hexamethylene bis acetamide (HMBA) and disulfiram compounds [113,114,115,116]. This also is an exciting target site since the P-TEFb catalyzes the phosphorylation of several transcriptional regulators at the HIV promoter site, which supports transcriptional initiation and elongation. Gliotoxin was previous reported as a virulent factor in *Aspergillus fumigatus* and showed immunosuppression properties [117]. It is a mycotoxin produced by species of *Aspergillus, Trichoderma* and *Penicillium* [118]. Furthermore, gliotoxin can be toxic when swallowed or inhaled and can cause skin and eye irritation when exposed. The oral dose of gliotoxin to be taken is 67 mg/kg [119]. These findings indicate fungal species as an untapped source of therapeutic agents that could expand treatment options for viral infections such as HIV-1. After the LRAs reactivate latent viruses, the immune system can eradicate the virus-producing cells. However, recent data indicated that CD8+ T cells in HIV-1-infected individuals on cART could not eliminate latently infected cells even after successful reactivation.

The currently available antiretroviral drugs target structural HIV-1 proteins but not HIV-1 accessory proteins (Nef, Vif, Vpu and Vpr). Accessory proteins in HIV-1 act as virulence factors mediating the severity of viral infection, replication, and disease progression. Although these proteins do not show any enzymatic activity, they act mainly with the host factors through protein–protein interactions [112]. Identifying natural products from microbial sources such as fungi could expand HIV-1 treatment options. Some advances have been made with fungal secondary metabolites inhibiting the HIV-1 viral protein R (Vpr). Fumagillin isolated from *Aspergillus fumigatus fresenius* was reported to inhibit the cell cycle arrest activity of Vpr in yeast and mammalian expression cells [115]. Fumagillin is a known mycotoxin that has been used as treatment for microsporidia fungal infection. It has been researched for cancer treatment by employing its angiogenesis inhibitory properties [115,116,117]. Kamata et al. [116] also showed that damnacanthal, a noni component, inhibits Vpr-associated cell death with no effect on the cell cycle. The advances in high-throughput screening platforms for antiviral compounds have the potential to accelerate the discovery of small molecular inhibitors that will expand to other accessory proteins.

While a few promising and successful efforts have been made in the early screening of fungi for anti-HIV-1 bioactive compounds, the full potential of the fungal kingdom has not been extensively explored to unearth its antiviral activities. Secondary metabolites isolated from *Alternaria alternata* have revealed the potential to inhibit HIV-1 at distinct stages of its life cycle; however, the compound responsible for the antiviral activity was not reported [49]. However, there are few reports regarding the mechanisms of action of these compounds. Therefore, further research is necessary to identify the mechanisms of these beneficial compounds isolated from endophytic/fungi with low cytotoxicity. More anti-HIV agents have been discovered from fungal species, placing the fungal kingdom as an exciting source for anti-HIV compounds. Table 1 summarizes fungal secondary metabolites that exhibit anti-HIV activities, inhibiting different stages of the HIV life cycle. Considering that natural products are structurally complex and may present with toxicity that could prevent their further development into viable drugs, the structural information of bioactive compounds could serve as a starting point in synthesis or semi-synthesis programs for the generation of analogues. This strategy has been successful in providing a sustainable supply of drugs inspired by naturally occurring bioactive structures and ensured that they are available in sufficient quantities [118,120].

Screening small compounds from natural sources, such as fungi, for antiviral (anti-HIV) bioactivity is still in its infancy. Nevertheless, the promising discoveries made thus far reveal the potential for further development of these bioactive compounds as lead compounds for HIV treatment. However, there are several limitations to the small molecular screens for anti-HIV compounds [48,121]. These obstacles include the need for highly specialized containment laboratories with highly trained personnel and a lack of innovative antiviral screening tools [48].

**Table 1 viruses-15-01039-t001:** Anti-HIV-1 compounds isolated from fungal species.

HIV-1 Life Cycle Stages	HIV-1 Agent/Compound	Structure	Fungal Species	Mechanism of Action	Activity Concentration	Reference
Fusion or entry inhibitors	Isochromophilones I C_23_H_25_O_5_Cl Mw 416.9	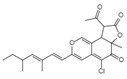	*Penicillium multicolor FO-2338*	Inhibited the gp120, CD4 binding	6.6 μM	[85,86,122]
	Isochromophilones II C_22_H_27_O_4_Cl Mw 390.9	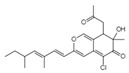			3.9 µM	
	Variecolin C_25_H_36_O_2_l Mw 368.55	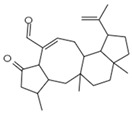	*Emericella aurantiobrumea*	Competed with macrophage inflammatory protein (MIP)-1 for binding to CCR5	9 µM	[87,122,123]
	Novel Sch 210972 C_25_H_35_NO_6_ Mw 445.55	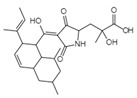	*Chaetomium globosum* Kunze SCF 1705	Strongly inhibited CCR-5 inhibitory activity	79 nM	[83,120,122,123]
	Penicillixanthone A C_32_H_30_O_14_ Mw 638.578	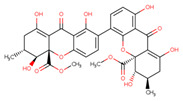	*Aspergillus fumigatus*	Inhibited CCR5-tropic HIV-1 SF162	0.36 μM	[123,124]
				Inhibited CXCR4-tropic HIV-1 NL4-3	0.26 μM	
Reverse Transcription	Stachybotrin D C_26_H_35_NO_5_ Mw 441.56	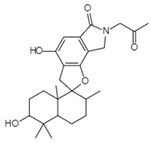	*Stachybotrys chartarum*	Inhibited the RT RNA-dependent DNA polymerase activity in a dose-dependent manner	8.4 µM	[122,125]
	Helotialins A C_23_H_30_O_6_ Mw 402.48	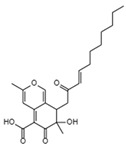	*Helotialean Ascomycete*	Not identified	8.01 nM	[122,126]
	Helotialins B C_22_H_30_O_4_ Mw 358.47	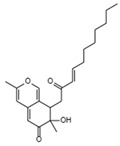			27.9 nM	
	Altertoxin Ⅴ C_20_H_14_O_6_ Mw 350.32	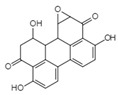	*Alternaria tenuissima*	Not identified	≤2.20 µM	[90,122]
Integration	Xanthoviridicatins E and F C_27_H_20_O_9_ Mw 488.44	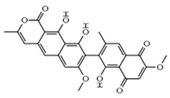	*Penicillium chrysogenum*	They inhibited the cleavage reaction of HIV-1 integrase.	6 and 5 μM	[93,122]
	Aquastatin A C_36_H_52_O_12_ MW 676.79	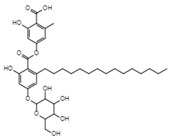	*Fusarium aquaeductuum*	Moderately inhibited the strand transfer reaction of HIV-1 integrase	50 µM	[122,127]
	Integracins B C_35_H_54_O_7_ Mw 586.8	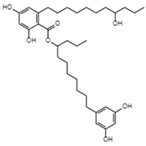	*Cytonaema* sp.	Inhibited coupled reaction of recombinant HIV-1 integrase	6.1 µM	[122,128]
	Integric acid C_25_H_52_O_6_ Mw 430.53	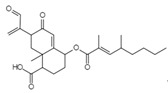	*Xylaria feejeensis*	Inhibited the 3′ end processing, strand transfer and disintegration	5–10 μM	[122,129]
	Atrovenetinone methyl acetal C_20_H_20_O_7_ Mw 372.37	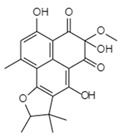	*Penicillium* sp. *FKI-1463*	Not identified	19 µM	[97,122]
Translation or budding	Hinnuliquinone C_32_H_30_N_2_O_4_ Mw 506.59	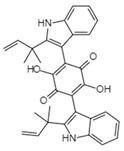	Unidentified fungus	Not identified	2.5 µM.	[100,122]
	Mulberroside C C_24_H_26_O_9_ Mw 458.46	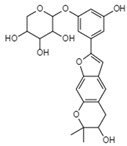	Unidentified fungus	Inhibited HIV-1 protease in a dose-dependent manner	<7.8 µg/mL	[56,122]

Structures presented in Table 1 were adapted from The Natural Product databases: Coconut [122] or Natural Product Atlas [123].

## 10. Conclusions

The recent progress in identifying antiviral compounds have potential to yield novel HIV therapeutics that can escape current treatment barriers. The recent genomic evolution has shown that fungal species holds a much greater biosynthetic potential with the revelation of several biosynthetic genes which are cryptic or transiently expressed under laboratory conditions. Here, we have shown that there is a concerted effort in developing unbiased antiviral screening assays. The challenge that limit early discovery efforts in identifying hit-to-lead compounds is the access to standardized antiviral screening systems that can map the inhibition impact of these antiviral compounds against various HIV target sites, including the HIV-1 proteins, cellular pathways, host factors and protein–protein interactions [59]. This suggests a need to standardize and validate currently existing antiviral screening assays and develop more comprehensive and high-throughput whole replication antiviral screening platforms to fully exploit the diverse antiviral mechanisms of small metabolites such as those produced by fungal species. Furthermore, the success of fungal derived antiviral compounds will only be realized if the hit compounds proceed from early drug development phases to advanced phases such as clinical trials. This will require a collaborative investment from research initiatives and Government sectors and pharmaceutical companies.

## Figures and Tables

**Figure 1 viruses-15-01039-f001:**
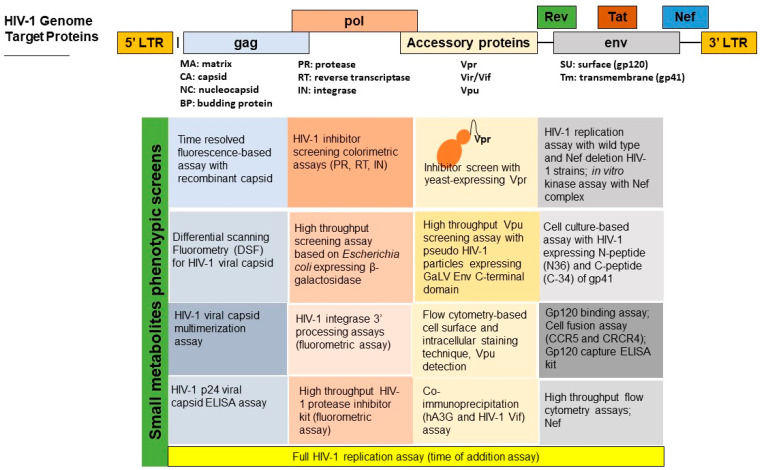
A summary of medium and high throughput screening assays for small molecular metabolites with anti-HIV-1 activity [54,55,56,57,58,59,60,61,62,63,64,65,66].

## Data Availability

Not applicable.

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
