# Peer review of "Developments in Exploring Fungal Secondary Metabolites as Antiviral Compounds and Advances in HIV-1 Inhibitor Screening Assays"

_viruses, 2023, doi:10.3390/v15051039_

Round 1

Reviewer 1 Report

my comments and suggestions are listed in the attached pdf "VIRUSES_2023_report".

Author Response

Reviewer 1

The third part is about the “current therapies in the clinical pipeline and under development”. The brief description of the different class (AZT, NRT, NNRT…) is common knowledge. It could be interesting to explicate a bit what is the biological target in the viral cycle of each of these classes. It could be interesting to discuss a bit the position of the available drugs in South Africa among the available drugs worldwide. Are they cost-effective? Are they responsible for the observed side effects? Are they other drugs that could be better tolerated? If yes, why are they not used? Is it because of the cost of such therapy? Why focus on South Africa (it does not seem to be the topic of the review)?

Responses:

  1. . It could be interesting to explicate a bit what is the biological target in the viral cycle of each of these classes.

Answer: We note that the reviewer already identified that the section discussing the current therapies need not to be extensive as it is common knowledge. Our aim of including a brief overview of this section is to lay a background on the available therapies. The naming of these drug classes is descriptive and suggest a target site, e.g., nucleoside reverse transcriptase inhibitors (NRTI) target the reverse transcription. Therefore, we hold a view that it might not be necessary to expand this section by adding the information on target sites.

  1. It could be interesting to discuss a bit the position of the available drugs in South Africa among the available drugs worldwide

Answer: The review is not only focused on the developments in South Africa, although we quoted the impact of available HIV-1 therapies in the Southern African region as this region is disproportionately impacted with HIV-1 infections and the largest rollout of ARVs compared to the global scenario. Also, the available drugs in South Africa are globally available, not region or country specific.

  1. Are they cost-effective?

Answer: There is limited information regarding the cost of these drugs and how they are procured. Therefore, the aim of this review was only to highlight the shortcomings of ARVs as a result of resistance development, non-adherence and inability to eliminate the proviral particles latently infecting the cells.

  1. Are they responsible for the observed side effects?

Answer: The side effects to any medicine depends on several factors such as the biochemistry of an individual, concentration used and tolerance to name a few. So, the side effects differ from one patient and can lead to non-adherence from patients, thus leading to development of resistance.

  1. Are they other drugs that could be better tolerated?

Answer: There is constant research on improving the available drugs individually or in combination. However, it was not in the interest of this review to detail the research progress in addressing this problem. The physicians normally do a medical assessment on the patient, if they record extreme side effects or that the drug combination prescribed for that patient is non-responsive, they typically prescribe another combination.

  1. If yes, why are they not used?

Answer: Kindly see our response above

  1. Is it because of the cost of such therapy?

Answer: In our setting, the cost might be a problem for Government as the treatment is provided for free to all qualifying patients. At a community level, it is not much about cost, but the effectiveness of the therapeutic intervention.

  1. Why focus on South Africa (it does not seem to be the topic of the review)?

Answer: In the review, we highlighted the Southern Africa region as this region is disproportionately affected compared to the rest of the world. We have deleted the sentence that specifically talk about the antiretroviral regimen available in South Africa as the reviewer has correctly pointed out that this is not the focus or the intention of the review.

It could be interesting to be more focused on the standard drugs associations, and explain why, despite the association, resistance is still a concern

Answer: During the drug combination in the first line treatment of HIV-1, different drugs that are targeting the different regions of HIV-1 replication are combined. First line regimen that is currently in use is the combination of dolutegravir, lamivudine and tenofovir disoproxil fumerate. This is the combination of integrase and reverse transcriptase inhibitors. Despite the combination therapy there is still development of HIV-1 drug resistance and also the combination is not effective in eradicating the virus in the reservoirs.

L95 to153: this part could be shortened and more specific about current strategies against HIV infections. Thus, this part could be merged with the precedent part about viral cycle

Answer: The sections on the HIV biology and current treatment strategies have been merged as “The biology of HIV and currently available therapies for HIV treatment” and has been shortened as suggested by the reviewer

L171 to 180: explicate a bit more the condition of the tests. In vitro I guess? Maybe, some reference data to compare might be interesting (what is the value of the positive control during these assays?)

Answer: We thank the reviewer comment. We have now indicated that these were from in vitro studies. These are early discovery studies. Unfortunately, in these studies no controls were reported.

This second part might be reorganized to be more understandable. For example by making a separate paragraph (with a title) for each supposed mechanism of action, then treating each level of development (in vitro, preclinical,…) to be more easy to read and by explicitly indicating when a drug is from fungal origin or not.

Answer: We thank the reviewer for this suggestion. The part referred to has been organized into subheadings as follows:

Fungal compounds as HIV-1 cell-surface receptor attachment inhibitors

Fungal compounds as HIV-1 reverse transcriptase inhibitors

Fungal compounds as HIV-1 integrase inhibitors

Fungal compounds as HIV-1 protease inhibitors

We have declared under the heading “Advances in developing fungal-derived small molecular inhibitors for major HIV-1 target sites” that these fungal metabolites are at the early stages of drug discovery (hit identification). We also highlight this under each subheading and emphasize the need for further development.

We have reviewed the subsections and removed all references made to compounds that are not of fungal origin.

Fungi or endophytic fungi? Please, be more specific. Some of the fungi of the Table are not endophytic.

Answer: We have removed indicators that the Table reports secondary metabolites from fungi and looked at fungi in general.

Ic could be interesting not to restrain to endophytic fungi since some of the presented fungi are not. But doing this, I think, the authors should complete their review with more specific examples, and, if it in the case, explain if the potential is only proof of concept (in vitro assay) or if more develop state is on (clinical assay, for example). The review lacks discussion on the potential of fungal metabolites in this context, which is a shame because it is the purpose of the review.

Answer: We have revised the title, abstract and introduction to indicate that we will be presenting fungal metabolites (not specifically from endophytic fungi)

L221 to 236: The “new” drugs mentioned are not so new, and not from fungal origins. Please precise this point.

Answer: We thank the reviewer for highlighting this, we have removed the statements referring to these drugs since they are not from fungi.

The authors do not explain how fungal metabolites have been tested. Is it fungal extracts, or plant tissues extract (since they are endophytes)? How have they been extracted (what method, what solvent)? Is it pure molecules? Please explain more

Answer: In this section, we have indicated that these are in vitro studies. We have also removed all compounds that are not of fungal origin. The compounds reported are all from fungi, not plants and in cases where a name for the molecule is reported, that represents a purified compound.

Since there is no standardized extraction method, we hold a view that if we present all different extraction methods and solvents used in each study, it will make the section complicated and not easy to read.

L303-304: this sentence should be mentioned in the beginning of the document. As the strategy on fungal metabolite mining strategies

Answer: The statement has been moved to the section: “Advances in developing fungal-derived small molecular inhibitors for major HIV-1 target sites”

The difference between the third part and the forth part is not clear (“current therapies in the clinical pipeline and under development” and “emerging HIV treatment strategies”)

Answer: The third part has been revised to: Current therapies for HIV inhibition in the clinical pipeline. Here we aim to lay the background on the currently available anti-HIV-1 therapies. The fourth part has been revised as the following subsections:

(1) Advances in developing fungal-derived small molecular inhibitors for major HIV-1 target sites and (2) Advances in developing fungal-derived small molecular inhibitors for new HIV-1 target sites

L330 to 342: are they from fungal origin? Please be specific

Answer: Thank you for noting this, the information where compounds not derived from fungi were mentioned has been removed.

In the paragraph “treatment strategies targeting latently infected cells” talking about fungal metabolites? If not, what is the purpose?

Answer: Compounds not produced by fungi has been removed and only the section that discuss fungal compounds with latency reversing properties was retained.

The paragraph “advances in antiviral screening assays” should be at the beginning of the review and shorten

Answer: The section has been moved to appear after “The biology of HIV and currently available therapies for HIV treatment” and have been revised to discuss specific assays

There is no conclusion

Answer: The section named as “Challenges and future avenues” has been renamed to “Conclusion”

Globally, the part about known, or not fungal drugs should be shorten to focus on the part about fungal drugs, which should be the core of the work, and more detailed and explained and discussed.

Answer: The part about known therapies have been shortened and merged to the HIV biology (HIV life cycle) and where compounds which are not from fungal origin were mentioned, the statements are now removed

Many of the cited papers are old, They are many references. Maybe a relevant and through selection could be made. It could be better than several old papers

Answer: We have made some efforts to remove some old articles, but we targeted articles reporting on structurally characterized compounds and with mechanism of action reported. Most of the recent studies report on crude extracts.

A lot of fungal metabolites are not presented, nor discussed. For example (not exhaustive)

Answer: We thank the reviewer for the suggested literature, we have considered some new studies (e.g., Tan et al. 2017) where they report on a chemically characterized compound as an entry inhibitor. We structured the review to focus on chemically purified compounds with anti-HIV activities.

The authors are focusing on HIV-1 but do not explain why.

Answer: There are two main HIV clades (HIV-1 and HIV-2), however, it is well documented that HIV-1 is a major focus as it leads to disproportionate mortalities, especially in highly impacted regions such as the Sub-Saharan Africa.

Show more details epidemiologic data and also on the side effects (which are one of your main argument that justify the need of new drugs), if known.

Answer: We have made a general discussion on the side effects of the available antiretroviral drugs as the epidemiological data reported differs from country to country. For this kind of analysis, it would require a review focusing on epidemiology and a qualified epidemiologist to analyze this data which is not in the scope of this review.

The term reservoir is used several times, in different contexts, and it should be more explained.

Answer: In line 371-378, under the subheading “Advances in developing fungal-derived small molecular inhibitors for new HIV-1 target sites” we have explained the meaning of the HIV reservoir as latently infected cells containing stably integrated, replication-competent HIV proviruses repressed at transcriptional and post-transcriptional levels by many silencing mechanisms. Furthermore, we detailed which cells are typical reservoirs of this latently integrated HIV proviral DNA (macrophages and microglia)

Please be careful: species names should be written in italics (some examples below, not exhaustive)

Answer: We thank the reviewer for noting this, we have corrected all scientific names are now in italics.

L35: this sentence is not clear, please explicite more

Answer: We have revised the statement to “However, these antiretroviral drugs do not cure HIV due to failure to eliminate latently integrated HIV proviral DNA in the host genome, which is capable of initiating repli-cation to detectable levels once ARV treatment is interrupted” in Line 36-39.

L45: please, be more specific about the reservoirs

Answer: Please see our response above, a detailed explanation of what is an HIV reservoir is in Line 371-378 under “Advances in developing fungal-derived small molecular inhibitors for new HIV-1 target sites”

L100: species name should be in italic

Answer: All species names are now in italics

L112: “new class” in 2007… 16 years ago is not new…

Answer: We have removed the statement in L 112

L173: Chaetomium globosum in italics

Answer: All species names are now in italics

L179: Penicillium nicillium in italics

Answer: All species names are now in italics

L182: Emericella aurantiobrunnea italics

Answer: All species names are now in italics

L203: Stachybotrys chartarum italics

Answer: All species names are now in italics

L206: Alternaria tenuissima italic

Answer: All species names are now in italics

L238: Penicillium chrysogenum italics

Answer: All species names are now in italics

Reviewer 2 Report

Reviewer comments

Reference:       viruses-2297801

Authors:    Bruce Nzimande, John P. Makhwitine, Nompumelelo P. Mkhwanazi, Sizwe I. Ndlovu

Tittle:               Endophytic fungi as emerging sources of antiviral compounds for HIV inhibition and immunomodulation

In this review, the research team led by N. P. Mkhwanazi and S. I. Ndlovu is intended to highlight the role of naturally-occurring compounds isolated from endophytic fungi in the treatment of HIV-1 infection. After a brief and well-reasoned introduction (29-75), the authors briefly review the key steps of the replication cycle of the pathogen (76-92) and, in a much more detailed fashion, describe the different families of agents approved for clinical use (93-118), the hurdles the current chemotherapeutic regimes face up (119-153) to finally emphasise the challenge that still represents the infection by HIV-1 despite the available chemotherapeutic arsenal to date and the importance of the development of new antiretroviral agents. The manuscript describes subsequently the main contributions in the field (154-310) from small-molecules isolated from fungi, data finally summarized in Table 1.

Many review articles have been published in recent time on the role and potential of natural products (from diverse, but also from fungal origin) in the context of the fight against AIDS. This manuscript is thematically focused on those produced particularly by filamentous fungi endowed of immunomodulatory and/or antiretroviral activities in the context of HIV-1 infection and, consequently, it is therefore highly specific. This is probably the main weakness of the manuscript: its extreme specificity when addressing this task.

However, the manuscript is further extended to the analysis of antiviral strategies targeting cellular pathways via protein-protein interactions (311-353), latently infected cells (354-433), HIV-1 accessory proteins (434-453) and novel antiviral screenings (454-564). Although fungal metabolites are cited, the aim and scope of these sections clearly exceeds by far the narrower original subject of the review and it is converted in a general review with a broader scope.  Very surprisingly, this manuscript simultaneously brings together both natures: a devoted specific subject and an analysis on more general aspects.

The manuscript is correctly written, adequately referred by primary bibliographic sources (outstanding 183 references cited!) and properly described/discussed when required.

After a careful and critical reading, the present manuscript gathers, in the opinion of this reviewer, enough interest as to be considered for publication. However, there are still some objections for this reviewer exposed as follow:

1-. The title does nor realistically reflect the full content of the review. It could misleads the reader into thinking that only the role of fungal metabolites is considered. A change is suggested in order to incorporate the full content and scope of the article.

2-. Reconsider a rewriting changing the structure of the paper: anticipate firstly the general aspects (311-564) and further the specific contributions due to endophytic fungal metabolites (154-310).

3-. Figure 1 should include the structures of the natural products referred. This would help to provide (in a visual and direct way) the spectrum of structural variability of the metabolites and, in the eyes of an expert, even facilitate the localisation of similarities in compounds targeting the same viral stage.

Final considerations:

Globally, this contribution contains a remarkable effort in systematising aspects of general interest with specific contributions from a specific  group of naturally-occurring fungal metabolites. Although average in terms of contents and originality, this contribution can be considered interesting, properly written and described/discussed. Consequently, I think the proposed manuscript could be of interest for the specialised reader and deserves an opportunity to be considered for publication. As a result of all the above stated, if authors take into consideration the few suggestions proposed by the reviewer (listed above) and provide reasonable responses to the points issued, providing an improved version of the manuscript according to the reviewer’s comments, the editor could give the opportunity to the authors of a resubmission

Author Response

In this review, the research team led by N. P. Mkhwanazi and S. I. Ndlovu is intended to highlight the role of naturally-occurring compounds isolated from endophytic fungi in the treatment of HIV-1 infection. After a brief and well-reasoned introduction (29-75), the authors briefly review the key steps of the replication cycle of the pathogen (76-92) and, in a much more detailed fashion, describe the different families of agents approved for clinical use (93-118), the hurdles the current chemotherapeutic regimes face up (119-153) to finally emphasise the challenge that still represents the infection by HIV-1 despite the available chemotherapeutic arsenal to date and the importance of the development of new antiretroviral agents. The manuscript describes subsequently the main contributions in the field (154-310) from small-molecules isolated from fungi, data finally summarized in Table 1.

Many review articles have been published in recent time on the role and potential of natural products (from diverse, but also from fungal origin) in the context of the fight against AIDS. This manuscript is thematically focused on those produced particularly by filamentous fungi endowed of immunomodulatory and/or antiretroviral activities in the context of HIV-1 infection and, consequently, it is therefore highly specific. This is probably the main weakness of the manuscript: its extreme specificity when addressing this task.

However, the manuscript is further extended to the analysis of antiviral strategies targeting cellular pathways via protein-protein interactions (311-353), latently infected cells (354-433), HIV-1 accessory proteins (434-453) and novel antiviral screenings (454-564). Although fungal metabolites are cited, the aim and scope of these sections clearly exceeds by far the narrower original subject of the review and it is converted in a general review with a broader scope.  Very surprisingly, this manuscript simultaneously brings together both natures: a devoted specific subject and an analysis on more general aspects.

The manuscript is correctly written, adequately referred by primary bibliographic sources (outstanding 183 references cited!) and properly described/discussed when required.

After a careful and critical reading, the present manuscript gathers, in the opinion of this reviewer, enough interest as to be considered for publication. However, there are still some objections for this reviewer exposed as follow:

1-. The title does nor realistically reflect the full content of the review. It could misleads the reader into thinking that only the role of fungal metabolites is considered. A change is suggested in order to incorporate the full content and scope of the article.

Answer: The title has been changed to accommodate the full scope of the review as follows: “Developments in exploring fungal secondary metabolites as antiviral compounds and advances in HIV-1 inhibitor screening assays”

2-. Reconsider a rewriting changing the structure of the paper: anticipate firstly the general aspects (311-564) and further the specific contributions due to endophytic fungal metabolites (154-310).

Answer: We thank the reviewers for their considered advise on improving the quality of the paper. The main aim of the review is to highlight currently available therapies that are in the clinical pipeline for well-established target sites and then provide a comprehensive update on the fungal compounds that has been identified for these target sites. We have changed the structure of the paper by combining the sections of the biology of the HIV (life cycle) and current therapies available. This is now followed by the section where we discuss antiviral screening assays, and this section has been revised. We then discuss the potential of fungal secondary metabolites as HIV-1 inhibitors where we first look at the fungal compounds targeting the major HIV proteins. The last section on emerging target sites has been revised.

3-. Figure 1 should include the structures of the natural products referred. This would help to provide (in a visual and direct way) the spectrum of structural variability of the metabolites and, in the eyes of an expert, even facilitate the localisation of similarities in compounds targeting the same viral stage.

Answer: We thank the reviewer for the suggestion, and we also consider that this will enable the expert to localise similarities in compounds mentioned. Unfortunately, natural products are complex molecules and mostly novel. We do not have expertise in redrawing these structures and we do not permission from the authors of the references studies to reproduce their compounds.

Final considerations:

Globally, this contribution contains a remarkable effort in systematising aspects of general interest with specific contributions from a specific  group of naturally-occurring fungal metabolites. Although average in terms of contents and originality, this contribution can be considered interesting, properly written and described/discussed. Consequently, I think the proposed manuscript could be of interest for the specialised reader and deserves an opportunity to be considered for publication. As a result of all the above stated, if authors take into consideration the few suggestions proposed by the reviewer (listed above) and provide reasonable responses to the points issued, providing an improved version of the manuscript according to the reviewer’s comments, the editor could give the opportunity to the authors of a resubmission

Answer: We thank the reviewer for the positive feedback and we hope we have provided detailed responses to most of the comments and suggestion and that the revised manuscript has improved in quality and presentation.

Reviewer 3 Report

In general, the article makes a good impression. The introduction rather extensively describes the existing problem of HIV infection and the difficulties associated with the development of new antiretroviral drugs. However, some text fragments are located chaotically and require redistribution. It should also be noted that some substances included in the review were isolated not from endophytic fungi, but from plants or algae, this point needs to be clarified.

A general note to the text is that the Latin names of plants and fungi should be italicized.

Line 154 - It is recommended to make any subheading when moving from information about the features of modern anti-HIV therapy to the biological activity of compounds of fungal origin.

Line 182 - It is recommended to make the subheading "Inhibitors of viral attachment to the cell receptor"

Line 187 - Griffithsin is a compound isolated from red alga Griffithsia - algae are not endophytic fungi. Why is this compound mentioned in the text of the article? It is necessary to either delete information about griffithsin or justify the need for its presence.

Line 195 - Recommended to be subtitled "Reverse Transcriptase Inhibitors"

Line 199-200 Several coumarins or their derivatives produced by plants and some fungal species - it is necessary to list which coumarins are meant. Do they have names?

Line 211. Helichrysum mimetes is a plant, not a fungus. It is necessary to either remove information or justify the need for its presence

Line 220 Recommended to be subtitled "Integrase Inhibitors"

Line 259 Recommended subheading "Protease Inhibitors"

Table 1 would be more logical to move to the end of the article, before the “conclusion” section

The description of screening assays in the “Advances in antiviral screening assays” section is very detailed and interesting, but has little to do with the main topic of the article. It is recommended that this section be substantially shortened.

The title of the section “Challenges and future avenues” is recommended to be changed to “Conclusion”, although the reviewer does not insist on this change.

Author Response

In general, the article makes a good impression. The introduction rather extensively describes the existing problem of HIV infection and the difficulties associated with the development of new antiretroviral drugs. However, some text fragments are located chaotically and require redistribution. It should also be noted that some substances included in the review were isolated not from endophytic fungi, but from plants or algae, this point needs to be clarified.

A general note to the text is that the Latin names of plants and fungi should be italicized.

Answer: Thank you for the note, we have italicized all scientific names accordingly.

Line 154 - It is recommended to make any subheading when moving from information about the features of modern anti-HIV therapy to the biological activity of compounds of fungal origin.

Answer: We have included the subheadings to clearly mark the transition from modern anti-HIV therapies to the biological activity of compounds of fungal origin under the following subheadings:

  1. Current therapies for HIV-1 inhibition in the clinical pipeline and those under development
  2. Advances in developing fungal-derived small molecular inhibitors for major target sites
  3. Advances in developing fungal-derived small molecular inhibitors for new HIV-1 target sites

Line 182 - It is recommended to make the subheading "Inhibitors of viral attachment to the cell receptor"

Answer: We have included the subheading as “Fungal compounds as HIV-1 cell-surface receptor attachment inhibitors”. Also, we included subheadings for the other target sites (reverse transcriptase, integrase and protease).

Line 187 - Griffithsin is a compound isolated from red alga Griffithsia - algae are not endophytic fungi. Why is this compound mentioned in the text of the article? It is necessary to either delete information about griffithsin or justify the need for its presence.

Answer: We thank the reviewer for noting this, we have deleted the information of griffithsin.

Line 195 - Recommended to be subtitled "Reverse Transcriptase Inhibitors"

Answer: We have now included the subheading as “Fungal compounds as reverse transcriptase inhibitors”. Similar subheadings have been included for other target sites.

Line 199-200 Several coumarins or their derivatives produced by plants and some fungal species - it is necessary to list which coumarins are meant. Do they have names?

Answer: The statement referring to several coumarins have been replaced with “

Also, Govindappa et al. [90] reported total coumarins from the crude extract of an en-dophytic fungus, Alternaria species to inhibit reverse trascriptase enzyme in a single protein in vitro colorimetric assay. The limit to this study was that they only reported total coumarins and no purification of the active compounds was reported”

Line 211. Helichrysum mimetes is a plant, not a fungus. It is necessary to either remove information or justify the need for its presence

Answer: We thank the reviewer for this comment, the information on H. mimetes has been removed as it falls outside the scope of this review.

Line 220 Recommended to be subtitled "Integrase Inhibitors"

Answer: The subtitle has been added as “Fungal compounds as integrase inhibitors”

Line 259 Recommended subheading "Protease Inhibitors"

Answer: The subheading has been added as “Fungal compounds as protease inhibitors”

Table 1 would be more logical to move to the end of the article, before the “conclusion” section

Answer: Table 1 has been moved to appear at the end as suggested by the reviewer.

The description of screening assays in the “Advances in antiviral screening assays” section is very detailed and interesting but has little to do with the main topic of the article. It is recommended that this section be substantially shortened.

Answer: We thank the reviewers for this well considered comment. We have decided to rather change the title to “Developments in exploring fungal secondary metabolites as antiviral compounds and advances in HIV-1 inhibitor screening assays” to expand the scope and focus of the review to the development of antiviral screening assays.

The title of the section “Challenges and future avenues” is recommended to be changed to “Conclusion”, although the reviewer does not insist on this change.

Answer: We have changed the section heading to “Conclusions” as suggested by the reviewer

Round 2

Reviewer 1 Report

Thanks for the work you have done on the manuscript and answers.

I kindly suggest few more little modifications.

- There are still several mentions indicating that you are focused on endophytic fungi (in the abstract and in the introduction (L55), as in the conlusion and among the text). Maybe you could be more explicit saying that endophytic fungi are of particular interest in this context even though you will review all fungal metabolites.

- If you are focused on pure molecules only, I think it should be clearly mentionned (for example, in the title, or L23-24 in the introduction) because, as you said, there also are studies about exctracts (crude or not).

- I understand why you chose to focus on South Africa (and HIV-1), but it should be explicitly mentioned in the text. Even though, it is not really clear why focus on the South Africa for the epidemiological part. What i do not understand is the link between this epidemiological focus and the fungal metabolites. The fungi and the metabolites you mentionned are not South African.

It could be more clear for your readers to give a general aspect of the epidemiological situation worldwide (just a few word), and then take South Africa as a relevant example justifying the strong need in new drug development targeting HIV-1. It could also be the opportunity to remind to your readers that HIV-1 is the most concerning.

- in table 1, I think this is Aspergillus fumigatus

- in Fig 1, what do mean the color gradient for each assay ? this should be explained.

- L278-279 and table 1 : you mentionned Penicillium nicillium. I think there is typing error. The reference you mentionned cited Penicillium sp. Here : https://pubmed.ncbi.nlm.nih.gov/7649871/ (Matsuzaki at al. 1995) they mentionned Penicillium multicolor (and it seems to be the same strain).

And here : https://pubmed.ncbi.nlm.nih.gov/29517908/ (luao et al. 2018), they mentionned different Isochromophilone from another fungi (Diaporthe sp.) but in a cyctotoxic context. A few word to discuss this point could be of interest.

- L387-395 : be careful, gliotoxin is not a new compound, and is also a well known mycotoxin that can be toxic. I think this point should be discussed a bit.

- L408-413 : same remark considering fumagillin, it is also a mycotoxin.

Author Response

  1. There are still several mentions indicating that you are focused on endophytic fungi (in the abstract and in the introduction (L55), as in the conclusion and among the text. Maybe you could be more explicit saying that endophytic fungi are of particular interest in this context even though you will review all fungal metabolites.

Response: Thank you for your comment. We have clarified that we will focus on all fungal metabolites although secondary metabolites from endophytic fungi are of interest. In the abstract, we have changed the statement in Line 16-18 as follows “The fungal species, more particular, fungal endophytes inhabiting medicinal plants' inner tissues, are gaining increasing attention as alternative sources of anti-HIV compounds or immunomodulators that can escape the current barriers to cure.” In the introduction we have changed the statement in Line 59-63 to read as follows “Fungal species, especially endophytic fungi associated with medicinal plants are rich sources of novel and bioactive small molecular (secondary metabolites) compounds [12-15]. Although this review will focus all fungal metabolites, endophytic fungi are of the particular interest since they represent a largely unexplored niche”.

  1. If you are focused on pure molecules only, I think it should be clearly mentioned (for example, in the title, or L23-L24 in the introduction) because, as you said, there also are studies about extracts (crude or not).

Response: Thank you for the comments. In this review we intended to review fungal secondary metabolites that have been characterized to elucidate the structure of compound exhibiting anti-HIV-1 activities. We have thus indicated this in the abstract (Line 23-26): In this, we first explore currently existing therapies for various HIV-1 target sites and then look at fungal secondary metabolites compounds that has been characterized to structural level and showing potential as inhibitors of various HIV-1 target sites. Also, in Line 67-70 in the Introduction we have indicated this intention: This review intends to highlight structurally elucidated compounds derived from fungi /endophytic fungi as alternative HIV treatment strategies that have the potential to expand HIV-1 treatment options and escape existing treatment barriers.

  1. I understand why you chose to focus on South Africa (and HIV-1), but it should be explicitly mentioned in the text. Even though, it is not really clear why focus on the South Africa for the epidemiological part. What I do not understand is the link between this epidemiological focus and the fungal metabolites. The fungi and the metabolites you mentioned are not South African.

Response: Thank you for your comments, I agree the problem of HIV is the globally problem. However, we have now only highlighted the Eastern and the Sub-Saharan Africa in the statistics as these regions are disproportionately affected by HIV-1. We have detailed that while the existing therapeutic interventions have achieved significant success, they are limited by treatment barriers such as drugs side effects which often leads to non-adherence and the emergence of drug resistance HIV-1 strains, collectively resulting in treatment failure. With this review, we want to draw attention to fungal secondary metabolites as there are sparsely reported studies showing their potential as compounds that can be effective against HIV.

  1. It could be clearer for your readers to give a general aspect of the epidemiological situation worldwide (just a few word), and then take South Africa as a relevant example justifying the strong need in new drug development targeting HIV-1. It could also be the opportunity to remind to your readers that HIV-1 is the most concerning.

Response: Thank you for your comments. The epidemiological statistics of HIV-1 has been in incorporated in Line 33-line 38. “According to UNAIDS report, more than 38 million people worldwide were living with HIV in the year 2021. About 20.6 million of these people living with HIV were in the Eastern and Southern Africa regions where HIV-1 is the predominant strain [1]. Of the 38 million infected people, 28.7 million people accessing antiretroviral therapy [1]. Despite these gains, about 650 000 people in this period succumbed to AIDS related illnesses.

  1. in table 1, I think this is Aspergillus fumigatus

Response: Thank you for noting the typo error has been corrected.

  1. in Fig 1, what do mean the color gradient for each assay ? this should be explained.

Response: The color scheme on Fig. 1 (HIV-inhibitor screening assays) are chosen to correspond with the HIV-1 target protein group. The gradient has no meaning, but we chose it to create a contrast between different screening assays so it easier for the reader to navigate.

  1. L278-279 and Table 1: you mentioned Penicillium nicillium. I think there is typing error. The reference you mentioned cited Penicillium sp. Here: https://pubmed.ncbi.nlm.nih.gov/7649871/ (Matsuzaki at al. 1995) they mentioned Penicillium multicolor (and it seems to be the same strain).

Response: Thank you for the comment. Penicillium nicillium has been changed to Penicillium multicolor FO-2338

8. And here : https://pubmed.ncbi.nlm.nih.gov/29517908/ (luao et al. 2018), they mentionned different Isochromophilone from another fungi (Diaporthe sp.) but in a cyctotoxic context. A few word to discuss this point could be of interest.

Response: Thank you for your comments. It has been discussed in Line 291 to line 294

  1. L387-395: be careful, gliotoxin is not a new compound, and is also a well-known mycotoxin that can be toxic. I think this point should be discussed a bit.

Response: Thank you for the comment and it is true, gliotoxin is a known molecule that is already available commercially. The statement has been changed to read “Stoszko et al. [112] recently isolated gliotoxin from a fungal secondary metabolite library and showed this compound to exhibit latency reversing properties.

  1. L408-413: same remark considering fumagillin, it is also a mycotoxin.

Response: Thank you for comment, it has been noted in Line 428-429.

In addition, we added a statement in Line 446-452 “Considering that natural products are structurally complex and may present with toxicity that could prevent their further development into viable drugs, the structural in-formation of bioactive compounds could serve as a starting point in semi-synthesis for generation of analogues. This strategy has been successful in providing a sustainable supply of drugs inspired by naturally occurring bioactive structures and ensured that they are available in sufficient quantities [123]. Here, we are emphasizing that while the elucidated compounds in their natural form may not be an instant fit for treatment purpose, they can serve as a starting point for optimization and formulation of highly potent semi-synthetic or synthetic drugs.

Reviewer 2 Report

After a critical reading of the new manuscript provided by the authors, I acknowledge that the suggested points 1 and 2 have been taken into consideration and incorporated in the improved version of the manuscript by the authors.

Point 3, however, has not been implemented. The authors support this in the lack of the appropriate software and no experience, which preclude them of  drawingf the structures. Unfortunately, I have to disagree as at the present moment there is free and very effective software available to carry out this task (for example, the programme chemsketch, free for academic and personal use (https://www.acdlabs.com/resources/free-chemistry-software-apps/chemsketch-freeware/)).

From the structure of the natural products (obtained from the original bibliographic source), it is possible with this sort of free software to draw them and incorporate into table 1.

I still consider that this structural information is sufficiently relevant and should be included in the article, which would improve it substantially.

If this task continues to be hard to perform by the authors, the editorial office could help in editing the images of the structures. In any case, only if the editor considers that this point represents an undefeatable stumbling block, it would be accepted the omission of the structures.

Otherwise, the review article meets the requirements to be accepted after (minor changes).

Author Response

After a critical reading of the new manuscript provided by the authors, I acknowledge that the suggested points 1 and 2 have been taken into consideration and incorporated in the improved version of the manuscript by the authors.

Point 3, however, has not been implemented. The authors support this in the lack of the appropriate software and no experience, which preclude them of  drawingf the structures. Unfortunately, I have to disagree as at the present moment there is free and very effective software available to carry out this task (for example, the programme chemsketch, free for academic and personal use (https://www.acdlabs.com/resources/free-chemistry-software-apps/chemsketch-freeware/)).

From the structure of the natural products (obtained from the original bibliographic source), it is possible with this sort of free software to draw them and incorporate into table 1.

I still consider that this structural information is sufficiently relevant and should be included in the article, which would improve it substantially.

If this task continues to be hard to perform by the authors, the editorial office could help in editing the images of the structures. In any case, only if the editor considers that this point represents an undefeatable stumbling block, it would be accepted the omission of the structures.

Otherwise, the review article meets the requirements to be accepted after (minor changes).

Response: Thank you for challenging us to include the structural information to enhance the quality of the comprehensive information provided in Table 1. We have tried to include the structures for the reported compounds, their molecular formular and molecular weight. The structures were adapted from the Coconut natural compounds database and some from the Natural Product Atlas database. These sources are acknowledged and referenced accordingly.

Reviewer 3 Report

All comments of the reviewer were taken into account and corrected. The article can be published.

Author Response

We would like to thank the reviewer for investing time in this submission and in assisting the authors to enhance the quality of the manuscript.